# Steroidogenic Enzyme and Steroid Receptor Expression in the Equine Accessory Sex Glands

**DOI:** 10.3390/ani11082322

**Published:** 2021-08-06

**Authors:** Robyn E. Ellerbrock, Giorgia Podico, Kirsten E. Scoggin, Barry A. Ball, Mariano Carossino, Igor F. Canisso

**Affiliations:** 1Department of Veterinary Clinical Medicine, College of Veterinary Medicine, University of Illinois Urbana-Champaign, 1008 W Hazelwood Drive, Urbana, IL 61802, USA; robyn.ellerbrock@uga.edu (R.E.E.); gpodico@illinois.edu (G.P.); 2Maxwell H. Gluck Equine Research Center, University of Kentucky, 1400 Nicholasville Rd, Lexington, KY 40546, USA; kirsten.scoggin@uky.edu (K.E.S.); b.a.ball@uky.edu (B.A.B.); mcarossino1@lsu.edu (M.C.)

**Keywords:** stallion, fetus, prostate, ampulla, vesicular gland, bulbourethral gland

## Abstract

**Simple Summary:**

The accessory sex glands are responsible for producing seminal plasma, and thus play a vital role in reproduction and fertility. While the horse rarely suffers from some of the accessory sex gland diseases affecting other domesticated animals, glandular secretions can nonetheless affect semen quality and survival, and little is known about the effects of steroid hormones on glandular development and regulation. This study assessed the expression level and distribution of the steroid receptors AR, ESR1, and ESR2, and the steroidogenic enzymes 3ΒHSD, CYP17, and CYP19 in the equine accessory sex glands at various stages of life and demonstrated that sex steroid receptors are present in all equine glands throughout life. In contrast, steroidogenic enzymes were only weakly and variably expressed, suggesting that the accessory sex glands are not significant sites of steroidogenesis.

**Abstract:**

The expression pattern and distribution of sex steroid receptors and steroidogenic enzymes during development of the equine accessory sex glands has not previously been described. We hypothesized that equine steroidogenic enzyme and sex steroid receptor expression is dependent on reproductive status. Accessory sex glands were harvested from mature stallions, pre-pubertal colts, geldings, and fetuses. Expression of mRNA for estrogen receptor 1 (ESR1), estrogen receptor 2 (ESR2), androgen receptor (AR), 3β-Hydroxysteroid dehydrogenase/Δ5-4 isomerase (3βHSD), P450,17α hydroxylase, 17–20 lyase (CYP17), and aromatase (CYP19) were quantified by RT-PCR, and protein localization of AR, ER-α, ER-β, and 3βHSD were investigated by immunohistochemistry. Expression of AR, ESR2, CYP17, or CYP19 in the ampulla was not different across reproductive statuses (*p* > 0.1), while expression of ESR1 was higher in the ampulla of geldings and fetuses than those of stallions or colts (*p* < 0.05). AR, ESR1 and ESR2 expression were decreased in stallion vesicular glands compared to the fetus or gelding, while AR, ESR1, and CYP17 expression were decreased in the bulbourethral glands compared to other glands. ESR1 expression was increased in the prostate compared to the bulbourethral glands, and no differences were seen with CYP19 or 3β-HSD. In conclusion, sex steroid receptors are expressed in all equine male accessory sex glands in all stages of life, while the steroidogenic enzymes were weakly and variably expressed.

## 1. Introduction

Accessory sex glands are located in the pelvic inlet and are primarily responsible for producing seminal plasma [1]. The presence, size and shape of the glands vary among domesticated mammals, with all accessory sex glands present in pairs (ampulla, vesicular glands, and bulbourethral glands) or lobules (prostate) in the horse [1]. In equids, the ampulla is a prominent dilation of the vas deferens averaging 10–25 cm in length, 1–2 cm in width, and with a 2 mm luminal diameter [1,2]. During emission, sperm stored in the tail of epididymis is transported to the lumen of the ampulla via vas deferens, then forceful contractions of the ampulla during ejaculation evacuate their contents into the *colliculus seminalis* [1]. Vesicular glands are sac-like structures located caudal to the ampulla on either side of the pelvic urethra and are responsible for producing the gel fraction of equine semen [1]. The prostate is a bi-lobulated gland located alongside the pelvic urethra, caudal to the vesicular glands, and originates as an outgrowth of the embryonic urogenital sinus [1]. The final accessory sex glands, the bulbourethral glands, are located at the caudal end of the pelvic urethra, and are covered by the bulbourethral muscle [1]. 

While development and regulation of the accessory sex glands is well described in some species (e.g., dogs) frequently affected by disease of the accessory sex glands, little is known about the development and function of these glands in the horse, despite the important role these glands play in determining fertility in other species. Sex steroid production and modulation are known to play an important role in reproductive development and regulation of the male reproductive tract in mammals [3,4]. Androgens are important for testicular development and maintenance of spermatogenesis in the horse, and presumably also play a role in accessory sex gland development and function [3]. In humans, androgens are vital to prostatic growth and tissue homeostasis [4]. The androgen receptor (AR) is highly expressed in the epithelium of the normal mature canine prostate, and expression decreases in diseased glands [5]. Given that horses are not prone to prostatic disease [6], it is possible that androgens do not play a significant role in glandular proliferation in the horse.

Estrogen plays a variety of roles in the male reproductive tract, from the regulation of resorption of luminal fluid in the efferent ducts [7] to the regulation of protein secretions [7,8,9]. Of interest, immunolocalization of ER-α and ER-β in the reproductive tract differs by species and tissue [7]. For example, while ER-β is found in all three epididymal regions throughout development in stallions, ER-α is less present in the epididymis of pre-pubertal animals and more prominent in post-pubertal animals [10]. This change corresponds with an increase in estradiol and suggests ER-α may be important in maturation of the reproductive tract at puberty. The relative expression of ER-α and ER-β receptors in the stallion accessory sex glands is currently unknown. 

Estrogens are often produced in peripheral tissues that express aromatase, and it is possible that the stallion accessory sex glands could be another site of estrogen or androgen production. 3β-Hydroxysteroid dehydrogenase/Δ5-4 isomerase (3βHSD) and P450c17 (CYP17) are two of the key enzymes regulating testosterone synthesis. Specifically, CYP17 converts 17α-hydroxyprengnenolone to dehydroepiandrosterone (DHEA), while 3βHSD catalyzes the conversion of Δ^5^ steroids such as DHEA to Δ^4^ keto steroids such as androstenedione, and aromatase is responsible for estrogen synthesis. In humans, expression of aromatase increases with age and prostatic diseases [4], while in horses it increases in Leydig cells with age [3]. Similarly, 3βHSD expression and localization changes with age in the male equine reproductive tract [3]. 3βHSD is highly expressed in the Leydig cells of mature stallions, and in the seminiferous tubules and Sertoli cells of prepubertal colts [3]. CYP17 expression is higher in adult stallion testes than prepubertal or pubertal testes, and immunolabeling localized CYP17 to Leydig cells in the equine testes [3]. Despite known presence of these enzymes in the equine reproductive tract, enzyme expression in the accessory sex glands of the horse is unknown. 

Furthering our understanding of the role that reproductive hormones play in the accessory sex glands could not only improve our knowledge of reproductive physiology, but also of subfertility in the stallion. In the present study, our objectives were to characterize the expression and localization of estrogen receptors, androgen receptor, and steroidogenic enzymes in the equine fetal, pre-pubertal colt, stallion, and gelding accessory sex glands. We hypothesized that steroid receptor and steroidogenic enzyme expression would vary based on life stage in the male equid, and that the accessory sex glands may be an additional site of steroid synthesis in the horse.

## 2. Materials and Methods

### 2.1. Tissue Collection 

Accessory sex glands were obtained from adult horses (*n* = 6) and fetuses (*n* = 3) euthanized for other research protocols at the University of Illinois of Illinois Veterinary Teaching Hospital, in Urbana IL, and from pre-pubertal animals (*n* = 3) euthanized at the Maine Chance Research Farm Department of Veterinary Science, University of Kentucky, Lexington, KY. Protocols were approved by the Institutional Animal Care and Use protocols at the University of Illinois (#1400, #14226, and #14243) and University of Kentucky (#2012-1046). Clinically healthy males were divided into the following groups: fetal (280–310 days gestation, *n* = 3), intact pre-pubertal (4, 8, or 10 months; *n* = 3), intact adult (4, 9, or 11 years; *n* = 3), and castrated adult (5, 8, or 13 years, *n* = 3). Accessory sex glands (vesicular gland, prostate, ampulla, and bulbourethral gland) were harvested immediately after euthanasia. Adult and pre-pubertal animals were euthanized via intravenous administration of pentobarbital sodium overdose. Fetuses were collected after induced parturition for another study and were euthanized immediately after delivery via intra cardiac injection of pentobarbital sodium. After euthanasia and en-block dissection of the reproductive tract from each carcass, tissues were placed on ice until further processing. Reproductive tract samples were chopped into 0.25 cm pieces, preserved in RNAlater (Invitrogen, Carlsbad, CA, USA), refrigerated overnight (4 °C), then frozen in RNAlater and stored at −80 °C until RNA isolation. Additional glandular samples were placed in 10% neutral buffered formalin for histological and immunohistochemistry analysis.

### 2.2. qPCR Analysis

The mRNA expression of androgen receptor (AR), estrogen receptor 1 (ESR1), estrogen receptor 2 (ESR2), 3β-Hydroxysteroid dehydrogenase/Δ5-4 isomerase (3βHSD), CYP17 (P450c17), and CYP19 (P450arom) were quantified by real-time quantitative PCR (qPCR) using primers shown in Table 1 as previously described [3,11]. Tissues were minced into small pieces using a scalpel blade, and then homogenized using a tissue homogenizer (Fisher Scientific, Arlington VT, USA). TRIzol Reagent (Invitrogen) was used to extract total cellular RNA from the samples according to the manufacturer’s recommendation. RNA was precipitated with isopropanol and 1/10 volume of 3 M sodium acetate, resuspended in deionized bi-distilled water, and then quantified by spectrophotometry (NanoDrop ND-1000; Agilent Technologies, Palo Alto, CA, USA). Glandular samples with a 260/280 ratio greater than 1.95, and a 260/230 ratio of at least 2.0 were used for analysis. For each reaction, 2 µg of RNA were then treated with rDNaseI (Invitrogen) for 30 min at 37 °C, followed by treatment with DNase Inactivation Reagent at room temperature for 2 min. A high-capacity cDNA reverse transcription kit and random hexamers (Invitrogen) were used for reverse transcription. The primers were designed using Primer-BLAST, and SYBR Green PCRMaster Mix (Invitrogen) was used for quantitative PCR. Cycling conditions were as follows: 95 °C for 10 min, 40 cycles of 95 °C for 15 s and 60 °C for 1 min, and 55–95 °C for dissociation, all reactions were performed in duplicate, and efficiencies were calculated using LinRegPCR (version 2012.0). 

Reactions were automatically pipetted by the epMotion Automated Pipetting Systems (Eppendorf, Westbury, NY, USA). Dissociation analysis at the end of each real-time run was used to verify the amplification of a single product and ensure specificity of amplification. All changes in gene expression were calculated by mean threshold cycle (C_T_) and then normalized for the housekeeping genes glyceraldehyde-3-phosphate dehydrogenase (GAPDH) and ribosomal 18s to generate delta (Δ) C_T_ values. All changes in relative abundance of transcripts were calculated by comparing the expression of the target transcript relative to the reference transcript Δ-C_T_ method [12]. 

### 2.3. Immunohistochemistry

Protein localization of AR, ER-α, ER-β, and 3βHSD in accessory sex glands were investigated by immunohistochemistry (IHC) using receptor and enzyme antibodies previously shown to be validated in the horse [3,10,13], as previously described [3,11]. Accessory sex gland sections (0.5–1 cm thickness) were fixed in 10% neutral buffered formalin for 24 h. After fixation, all samples were dehydrated and embedded in paraffin, and later sectioned in 5 µm slices for IHC. For immunostaining of tissues, a mouse anti-α-bovine ER-α monoclonal antibody (1:25, sc-787, Santa Cruz Biotechnology Inc., Santa Cruz, CA, USA), a mouse anti-human estrogen receptor beta (ER-β) 1 isoform monoclonal antibody (1:20, MCA1974S, AbD Serotec, Raleigh, NC, USA), a goat 3βHSD polyclonal antibody (1:500, SC-30820, Santa Cruz Biotechnology Inc., Santa Cruz, CA, USA), and a rabbit anti-human AR polyclonal antibody (1:1000, SC-816, Santa Cruz Biotechnology Inc., Santa Cruz, CA, USA) were selected.

Briefly, the slides were exposed to automated dewaxing and rehydration steps, and then antigen retrieval was conducted using heat (100 °C for 30 min) and a pH 6.0 citrate-based ready to use solution (Leica Biosystems, Wetzlar, Germany). The slides were subsequently incubated with 3% hydrogen peroxide for 5 min, then a diluted primary antibody for 15 min, followed by a rabbit anti-mouse IgG for 8 min, a polymer-labeled goat anti-rabbit IgG coupled with horseradish peroxidase for 8 min, and finally a diaminobenzidine substrate for 10 min with reagents from the Bond Polymer Refine Detection system (Leica Biosystems). Primary antibodies were diluted to the desired concentration using Bond Primary Antibody Diluent (Leica Biosystems). The washing steps between each reagent were performed using Bond Wash solution 10× concentrate (Leica Biosystems) diluted to a 1× working solution with distilled water. After staining, serial ethanol dilutions were used to dehydrate all slides, and slides were then immersed in four baths of 100% xylene and mounted for analysis. Slides were viewed at 40× and 100× magnification for analysis.

### 2.4. Histology

Five-micron sections of formalin-fixed, paraffin embedded (FFPE) tissue derived from all glands of each horse at all life stages were stained with Masson’s Trichrome, hematoxylin, and eosin for structural evaluation following standard procedures. 

### 2.5. Quantitative Analysis of Immunohistochemistry 

For the quantification of AR, ER-α, ER-β, images were acquired from each section in ten different fields at 40× magnification (Uplan Apo 40×/0.85) with a camera (ProgRes C14plus, Jenoptik, Germany) coupled with a microscope (Olympus BX51, Olympus corporation, Tokyo, Japan). 

Images were analyzed using ImageJ (Version 1.52a, National Institute of Health, Bethesda, MD, USA) and IHC toolbox, a semi-automatic plugin of ImageJ [14]. The positive colors were manually selected in at least ten random images of each receptor. A statistical model for the color detection was created and applied to the other images during the analyses. In each image, pixels stained with the specific color were detected and extracted from the background. Subsequently, the HSB-threshold function of ImageJ was used to quantify the percentile of the area occupied by the positive staining. 

### 2.6. Statistical Analysis

Normal distribution of data was confirmed by the Shapiro–Wilk Test, and normal variance was confirmed with the Bartlett’s test. The ΔC_T_ values for each status and gland were analyzed by one-way ANOVA, and when significant, post hoc comparisons with Tukey’s HSD. Similarly, glandular immunolabeling intensity was analyzed by one-way ANOVA and when significant, post hoc comparisons with Tukey’s HSD. Significance was set at *p* < 0.05. Statistical analyses were carried out using R version 4.0.3 (10 October 2020) [15].

**Table 1 animals-11-02322-t001:** Primer sequences for real-time quantitative PCR of equine accessory sex glands [3,16].

Genes (Locus)	Primer Sequence	Ref.
AR (XM 01504865.1)	Forward: 5′-AGCTGCCATCCACTCTGTCT-3′	[11]
	Reverse: 3′-TGATAAACTGCTGCCTCGTC-5′	
ESR1 (NM 001081772)	Forward: 5′-TCCATGGAGCACCCAGGAAAGC-3′	[16]
	Reverse: 3′-CGGAGCCGAGATGACGTAGCC-3′	
ESR2 (XM 001915519)	Forward: 5′-TCCTGAATGCTGTGACCGAC-3′	[16]
	Reverse: 3′-GTGCCTGACGTGAGAAAGGA-3′	
HSD3B1 (D89666.1)	Forward: 5′-AGCAAATACCATGAGCACGA-3′	[3]
	Reverse: 3′-TAACGTGGGCATCTTGTGAA-5′	
CYP17 (GenBank# D30688.1)	Forward: 5′-GCATGCCTGGACTTACTGATCC-3′	[3]
	Reverse: 3′-CTGGGCCAGTGTTGTTATTG-5′	
CYP19 (Horse Genome #AF031520.1)	Forward: 5′-CCACATCATGAAACACGATCA-3′	[3]
	Reverse: 3′-TACTGCAACCCAAATGTGCT-5′	

## 3. Results

### 3.1. Histological Assessment of Accessory Sex Glands

The majority of the fetal ampulla and prostate was stromal, with limited small diameter ducts lined by few branched tubular glands. The colt prostate and ampulla had a higher epithelium to stromal ratio, and many sperm were noted in the lumen of the stallion ampulla. Branching of the tubulo-alveolar glands was maximal in the stallion, and very blunted with minimal branching in the gelding ampulla, as seen in Figure 1. In contrast, the tubulo-alveolar glands of the fetal bulbourethral gland were already developed, and changes between the life stages was minimal. Some tubular dilation was noted in the gelding bulbourethral gland. In contrast to the bulbourethral gland, the vesicular glands were mainly connective tissue in the fetus, colt, and gelding, and columnar glands were only extensive in the stallion, as can be seen in Figure 2. 

### 3.2. Androgen Receptor

AR mRNA expression did not differ between reproductive stages in the ampulla, bulbourethral, or prostate glands (*p* > 0.05), as can be seen in Figure 3. In the vesicular gland, AR expression was significantly higher in the fetus than in the stallion (*p* < 0.05). The bulbourethral glands showed the lowest level of AR mRNA expression compared to the other accessory glands. Similarly, the percentage of cells labeled by IHC, depicted in Table 2, did not differ between reproductive stages, but the percentage cells labeled were lower in the vesicular gland than the bulbourethral gland or prostate. 

The androgen receptor was strongly localized to the nuclei of the ampulla epithelium in all life stages, with weaker localization to the epithelial cytoplasm, as can be seen in Figure 4. There was marked localization to stromal cells of the fetal ampulla and colt ampulla, while stromal cell labeling of the gelding ampulla was sporadic, with labeling of some, but the majority of stromal cells. Interestingly, AR was strongly localized to the prostate epithelial cell nuclei in the fetus and stallion, and was largely absent from the nuclei of the colt and gelding epithelium. All four life stages had sporadic staining of prostate stromal cell nuclei. In the bulbourethral gland, AR was strongly localized to the epithelial cell nuclei in the fetus, colt, and stallion, however, only weakly present in the gelding epithelium. All life stages had intermittent immunolabeling of stromal cell nuclei in the bulbourethral gland. Similar to the ampulla, AR was strongly localized to the epithelial cell nuclei of the vesicular gland for all four life stages, had moderate intermittent labeling of stromal cell nuclei in the colt and stallion, and weaker intermittent labeling of fetal and gelding vesicular stroma. 

### 3.3. Estrogen Receptor 1

ESR1 mRNA expression was higher in the ampulla of geldings and fetuses than in colts and stallions, and lower in the stallion vesicular gland than in the other three reproductive stages (*p* < 0.05) as shown in Figure 5. No differences in expression were noted in the bulbourethral or prostate glands. Similar to AR, expression of ESR1 was lower in the bulbourethral gland than the other three glands.

In contrast, glandular intensity with ER-α immunolabeling was greatest in the bulbourethral gland, and lowest in the vesicular and prostate glands (Table 2). The percentage of cells with immunolabelling was higher in the stallion ampulla than colt or fetal ampulla, higher in the stallion prostate than the colt, and higher in the stallion vesicular gland than the colt, fetal, or gelding vesicular glands. ER-α was localized to the cytoplasm of the epithelium in all glands, with mild localization to smooth muscle cells, and sporadic labeling of stromal cell nuclei in the bulbourethral gland. 

### 3.4. Estrogen Receptor 2

ESR2 mRNA expression was lower in the stallion than the colt, fetus or gelding vesicular glands (*p* < 0.05) and did not vary between reproductive stages for the ampulla, bulbourethral, or prostate glands. Glandular expression was decreased in the prostate compared to the other glands (*p* < 0.05). With ER-β immunolabeling, the percent of positively labeled cells did not differ between reproductive statuses in the ampulla or bulbourethral glands, but was stronger in the stallion than colt prostate, and stronger in the stallion vesicular gland than other reproductive stages (Table 2). The percentage of cells that were immunolabeled was significantly higher in the ampulla and bulbourethral glands than the prostate or vesicular gland. 

Immunolabeling of ER-β was strong in the epithelial nuclei and cytoplasm of the stallion and gelding, and only moderate in the colt and fetus, as seen in Figure 6. In the bulbourethral gland, ER-β localized to epithelial nuclei in all life stages, with moderate localization to the epithelial cytoplasm in the stallion and gelding, and sporadic stromal cell labeling in all life stages. Prostatic and vesicular gland immunolabeling of ER-β was marked in epithelial nuclei, and moderate in epithelial cytoplasm for all life stages. There were intermittent stromal cell nuclei labeling. 

### 3.5. Steroidogenic Enzymes

Expression of 5α-reductase (CYP17, P450c17) was not different between life stages for any of the glands, and expression of 5α-reductase was minimal in the bulbourethral glands compared to the other three glands. 

Aromatase expression was similar between life stages in all glands, with the exception that aromatase expression was not detected in the stallion ampulla and colt bulbourethral glands. Additionally, expression of aromatase was significantly lower in the bulbourethral gland that the other glands when looking at all life stages combined (*p* < 0.05). 

3βHSD expression was low in all glands, and was similar between life stages, except for the gelding ampulla, where expression was undetectable. Expression was higher in the prostate than in the bulbourethral gland or vesicular gland (*p* < 0.05). 3βHSD immunolabeling was very weak to absent for all glands and all life stages. 

## 4. Discussion

The accessory sex glands play a vital, but underdiscussed, role in stallion fertility, and a better understanding of glandular development and regulation may help improve fertility in some animals. The objectives of this study were to determine if steroid receptors were present in all equine accessory sex glands, to determine if steroidogenic enzymes were present in the equine accessory sex glands, and to determine if receptor and enzyme expression differed based on age and castration status. As hypothesized, steroid receptors were detected in all accessory sex glands in all life stages in the horse, with variable changes in expression throughout development, suggesting that they play a role in accessory sex gland regulation and function. In contrast, CYP17, CYP19, and 3βHSD expression were weak to absent in all glands, suggesting that these glands are not a likely site of steroid synthesis in the horse. 

Androgen receptor mRNA expression varied among life stages and glands. There was increased expression in the fetal vesicular gland compared to the stallion, and lower expression in the bulbourethral gland as opposed to other glands. This pattern is consistent with other tissues in the stallion, such as the epididymis, where AR was present in all regions throughout development and adulthood [10]. Androgens regulate AR expression in other species [17,18], and maintain epididymal epithelial cell function and secretion [19]. The consistent expression of AR throughout the stallion accessory sex glands suggests a similar role in the stallion. The apparent lack of localization to the epithelium in the gelding prostate and bulbourethral gland suggests that a decrease in androgens in the castrated male may decrease AR expression in the luminal epithelium without affecting AR expression the stroma. Similar results have been demonstrated in the dog and human, where castration or withdrawal of hormone treatment lead to a decrease in epithelial and stromal AR signaling [20,21]. Dogs treated with antiandrogenic agents also have a marked decrease in prostatic AR expression [22,23]. In contrast to our findings in the prostate and bulbourethral gland, AR expression in the gelding ampulla and vesicular epithelium is similar to the stallion and appears to be independent of castration status. 

Both ER-α and ER-β were present in the accessory sex glands of the horse. In the current study, ER1 mRNA expression was higher in the gelding or fetal ampulla than the colt or stallion, and lower in the stallion vesicular gland than the other three glands. However, a higher percent of cells showed ER-α immunolabeling in the stallion ampulla than the colt or fetus, and localized to the epithelial cytoplasm of all glands, and all stages, suggesting that unlike the epididymis [10], ER-α may not play a role in the maturation of the accessory sex glands. This is in contrast to rats, where ER-α was not detected in the epithelial cells at any life stage, and was only faintly detected in the stroma of neonatal rats [24]. ER-α localization is variable in the equine epididymis after puberty, localizing to the caput in all animals, and additionally to the cauda in some animals [10]. ER-α and ER-β differential localization suggests regional and age-related regulator effects, such as ER-α being responsible for fluid resorption in efferent ducts (21). It has been suggested that ER-β may be more important for maintenance and homeostasis of epididymis, and ER-α may have region-specific effects on luminal fluid and protein secretion on other tissue [10], and similar effects may be possible in the equine accessory sex glands.

In the horse accessory sex glands, expression of ER-β did not vary between life stages for any gland besides the vesicular gland, and this difference was once again not seen when evaluating the percent of cells with positive immunolabeling. With immunolabeling, the percentage of cells positive for ER-β was increased in the stallion prostate and vesicular glands. Most interestingly, while overall percentage immunolabeling or mRNA expression did not change, localization of ER-β in the prostate was different in the stallion and fetus, compared to the colt and gelding, suggesting the potential for androgen regulation of ER-β expression in the equine prostate. In contrast, in rats, immunostaining of ER-β in the prostate epithelial nuclei increases in intensity with age [24].

While the role of ER-α and ER-β in accessory sex gland disease in the horse is not known; in humans, ER-α activation is associated with aberrant proliferation, inflammation and prostate cancer [4]. In contrast, ER-β activation appears to mediate beneficial anti-proliferative, anti-inflammatory, and potentially, anti-carcinogenic effects of estrogen [4]. Similarly, activation of the ER-β receptor in testosterone-treated mice prevents the expected prostatic hyperplasia and inflammation [25], and when ER-β in knocked out in the mice. In the current study, no clear difference was determined between ER-α and ER-β that might indicate a role in glandular regulation, but no animals in the current study were exposed to exogenous hormones. 

Aromatase expression increases in the stallion testis with age [26,27], and it was unexpected that we did not see a difference in the accessory sex glands with age. This may be in part due to the relatively low expression in the glands across all life stages and appears to indicate that the stallion accessory sex glands are not a significant source of estrogen synthesis and may be one reason that the stallion prostate is not prone to disease. Alternatively, it is possible that more sensitive methods may be needed to detect aromatase expression in the equine accessory sex glands. In humans, early studies did not detect aromatase in the prostate using biological assays [28,29], but later studies detected aromatase expression on RT-PCR, and detected enzymatic activity using alternative biochemical assays [30,31,32,33]. Laser capture microdissection and separation of prostate epithelial and stromal cells definitively demonstrated aromatase in the human prostate stromal cells, and a lack of aromatase expression in healthy prostatic epithelial cells, and it is possible similar methods might detect different results in the horse [34]. 

Finally, 3βHSD expression was almost undetectable in all accessory sex glands in all life stages of the male horse, also confirming that the accessory sex glands are not likely to be a site of significant androgen synthesis in the horse. While this is not unexpected, 3βHSD has been previously detected via immunohistochemistry in the human prostate [35], and remains to be explored in other species. 

## 5. Conclusions

In conclusion, AR, ER-α and ER-β are all present in the equine accessory sex glands, with variable changes in expression throughout development, suggesting that they play a role in accessory sex gland regulation and function. In contrast, lack of strong CYP17, CYP19, or 3βHSD expression suggests that the equine accessory sex glands do not play a significant role in steroid hormone production in the horse. 

## Figures and Tables

**Figure 1 animals-11-02322-f001:**
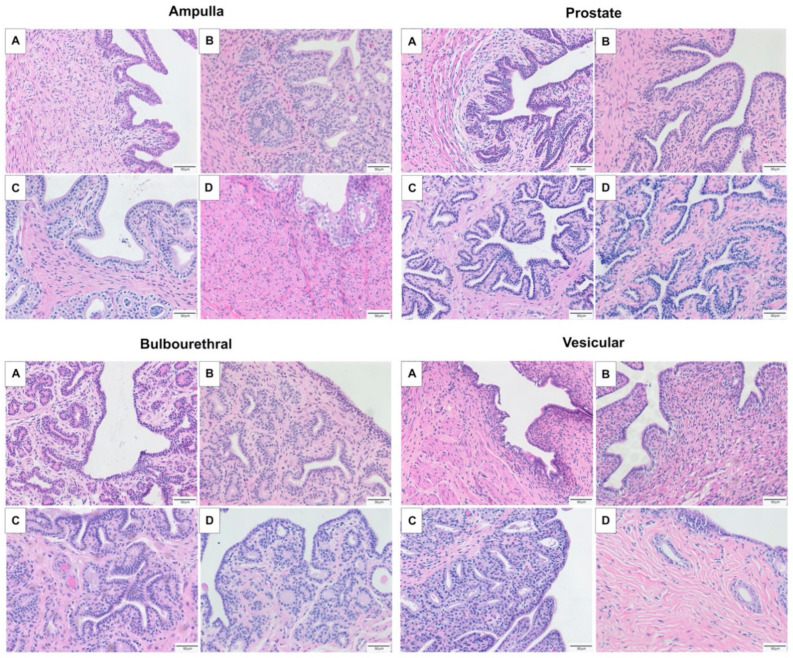
Representative images of the accessory sex glands of the fetus stained with hematoxylin and eosin (**A**) 280 days gestation, prepubertal colt (**B**) 4–10 months, mature stallion (**C**) >5 years, and gelding (**D**) >5 years.

**Figure 2 animals-11-02322-f002:**
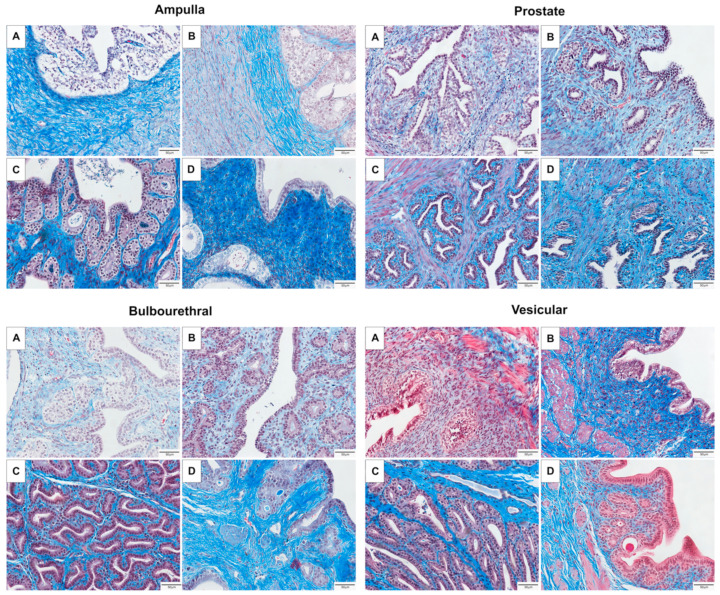
Representative images of the accessory sex glands stained with Masson’s Trichrome stain. Glands pictured come from a male fetus (**A**) 280–310 days gestation, prepubertal colt (**B**) 4–10 months, mature stallion (**C**) >5 years, and gelding (**D**) >5 years.

**Figure 3 animals-11-02322-f003:**
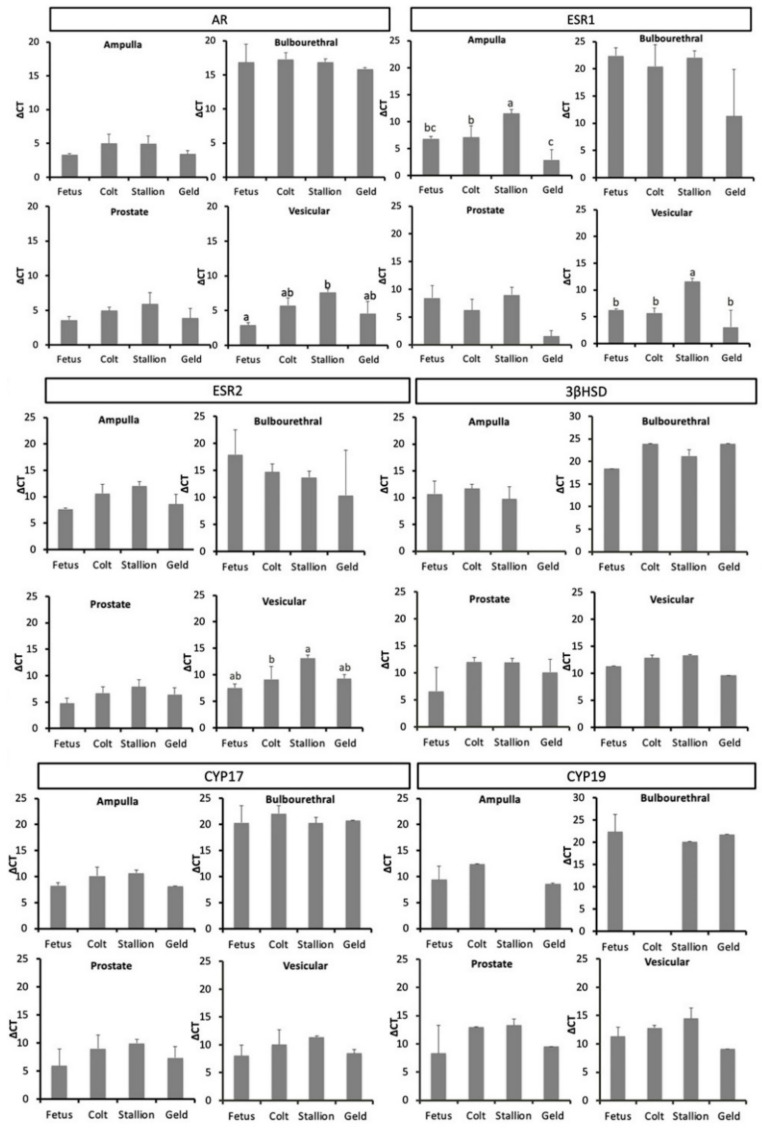
Mean mRNA expression in male equine accessory sex glands during different life stages (*n* = 3 per life stage). Data are expressed as mean ΔC_T_ relative to expression of GAPDH. Within glands, life stages with different superscripts differ in expression (*p* < 0.05). Abbreviations: AR, androgen receptor; ESR1, estrogen receptor 1; ESR2, estrogen receptor 2; 3βHSD, 3β-Hydroxysteroid dehydrogenase/Δ5-4 isomerase; CYP17, cytochrome P450 17α-hydroxylase/17,20 lyase; CYP19, P450-aromatase.

**Figure 4 animals-11-02322-f004:**
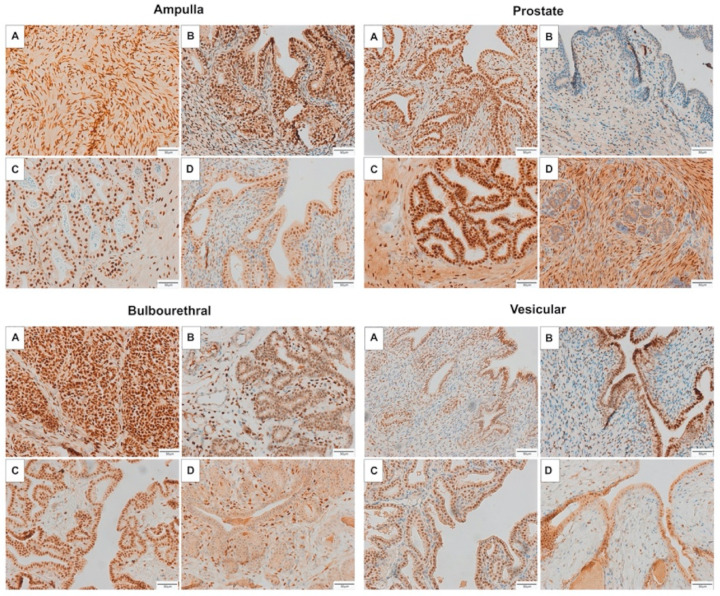
Immunolabeling of AR in the accessory sex glands of the fetus (**A**) 280–310 days gestation, prepubertal colt (**B**) 5–10 months, mature stallion (**C**) >5 years, and gelding (**D**) >5 years. AR localized to the epithelium of glands of all life stages except the colt and gelding prostate, and there was heterogenous localization to stromal cell nuclei.

**Figure 5 animals-11-02322-f005:**
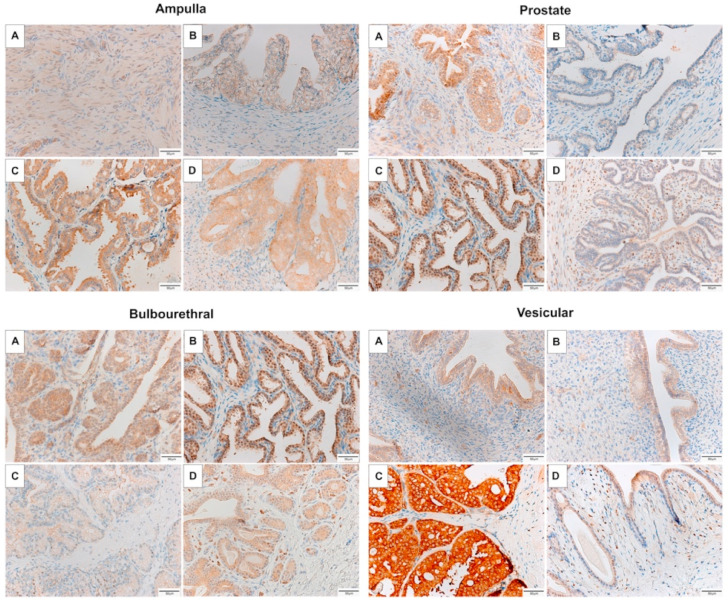
Immunolabeling of ER-α in the accessory sex glands of the fetus (**A**) 280–310 days gestation, prepubertal colt (**B**) 5–10 months, mature stallion (**C**) >5 years, and gelding (**D**) >5 years. ER-α localized to the epithelium of glands of all life stages.

**Figure 6 animals-11-02322-f006:**
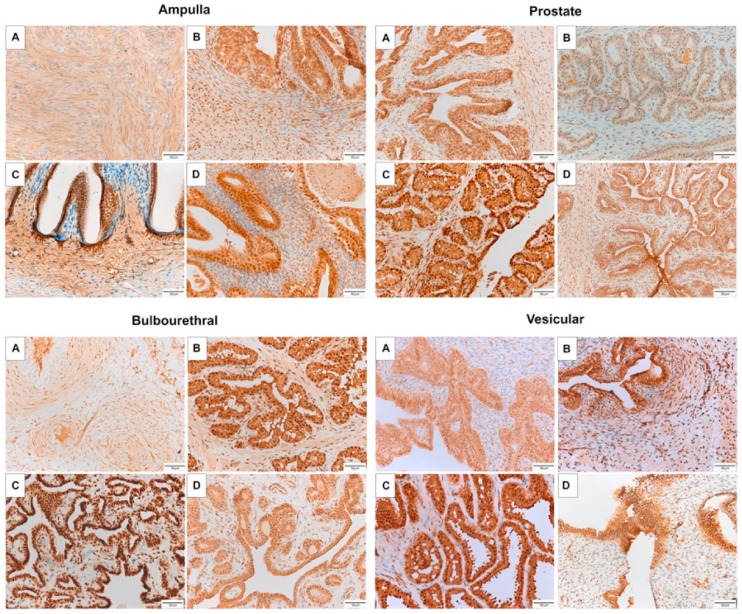
Immunolabeling of ER-β in the accessory sex glands of the fetus (**A**) 280–310 days gestation, prepubertal colt (**B**) 5–10 months, mature stallion (**C**) >5 years, and gelding (**D**) >5 years. ER-β localized to the epithelium of glands of all life stages, as well as to stromal cells.

**Table 2 animals-11-02322-t002:** Mean positive immunolabeling (%) for Androgen Receptor (AR), Estrogen receptor alpha (ER-α) and Estrogen receptor beta (ER-β) in male equine accessory sex glands of different life stages (fetus 280-310 days gestation, prepubertal colt 8-10 months, mature stallion > 5 years, and gelding > 5 years), *n* = 3 per life stage. Superscripts denote significant difference (*p* < 0.05) within that row.

AR				
	Fetus	Colt	Gelding	Stallion
Ampulla	73.9 ± 25.1 ^a^	63.6 ± 10.1 ^a^	64.8 ± 3.8 ^ab^	56.6 ± 3.3 ^b^
Bulbourethral	83.2 ± 9.9 ^a^	58.8 ± 11.1 ^b^	69.4 ± 0.6 ^b^	65.7 ± 13.0 ^ab^
Prostate	67.4 ± 18.7 ^a^	50.4 ± 10.6 ^b^	75.2 ± 17.3 ^a^	68.0 ± 13.7 ^a^
Vesicular	56.7 ± 8.6 ^ab^	48.5 ± 10.0 ^ab^	41.6 ± 25.9 ^a^	55.4 ± 4.6 ^b^
**ER-α**				
	**Fetus**	**Colt**	**Gelding**	**Stallion**
Ampulla	23.3 ± 8.7 ^ab^	21.3 ± 8.7 ^a^	28.8 ± 8.8 ^b^	46.5 ± 8.8 ^c^
Bulbourethral	35.0 ± 10.6	31.7 ± 21.2	38.5 ± 12.7	38.8 ± 19.2
Prostate	21.5 ± 11.9 ^a^	14.0 ± 8.6 ^a^	16.4 ± 6.0 ^a^	31.0 ± 4.2 ^b^
Vesicular	19.5 ± 4.9 ^b^	16.3 ± 5.0 ^b^	17.5 ± 5.8 ^b^	40.3 ± 11.3 ^a^
**ER-β**				
	**Fetus**	**Colt**	**Gelding**	**Stallion**
Ampulla	64.7 ± 23.5	63.3 ± 18.6	64.3 ± 2.4	67.2 ± 10.6
Bulbourethral	55.8 ± 3.4 ^a^	60.8 ± 7.1 ^b^	64.2 ± 3.8 ^b^	70.2 ± 9.7 ^b^
Prostate	44.0 ± 28.3 ^a^	43.2 ± 26.5 ^ab^	56.4 ± 15.4 ^b^	67.0 ± 17.5 ^c^
Vesicular	22.9 ± 13.1 ^a^	42.4 ± 15.6 ^a^	25.4 ± 16.0 ^a^	62.6 ± 16.7 ^b^

## Data Availability

Data sharing not applicable.

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
