# Peer review of "Steroidogenic Enzyme and Steroid Receptor Expression in the Equine Accessory Sex Glands"

_animals, 2021, doi:10.3390/ani11082322_

Round 1

Reviewer 1 Report

General comments: The manuscript entitled “Steroidogenic enzymes and steroid receptor expression in the accessory sex glands of the horse” (Manuscript ID; animals-1286059 ) was describes the expression level and distribution of steroids receptors and steroidogenic enzymes in the accessory sex glands of the horse evaluating by real-time PCR and immunohistochemistry. Results demonstrated that sex steroid receptors (AR, ER-a, ER-b) were expressed in all accessory sex glands throughout all stage of life in male horse, however the expression of steroidogenic enzymes were weak. The manuscript showed basic information for understanding of accessory glands regulation and development by steroid receptors. There are some points needing corrections in the manuscript. Please consider the suggested edits listed below.

Throughout the manuscript: Please consistent the abbreviation word of 3β-Hydroxysteroid dehydrogenase/Δ5-4 isomerase.

Abstract: I could not understand which results were mentioned about the results of mRNA or protein. Please clarify it.

L33: "ER1" to "ESR1"

Introduction and qPCR analysis: Please add the words of each abbreviation of gene and protein when each words appear at first.

L70: Please delete [7-9]. Tissue collection: In this study each groups' replicates were three. Therefore, please add the individual's month or age of each horse. Also, I wonder that are there any differences of expression level of steroid receptors among age in each group?

Statistical analysis: Did you analyze the normal distribution by Shapiro-Wilk test and homogeneity of variance by Bartlett's test of each data? If so, please describe it. When the normality and homogeneity of variance of data are rejected by Shapiro-Wilk test and/or Bartlett's test, you have to analyze the data by Kruskal-Wallis test.

Author Response

Responses to Reviewer #1

General comments: The manuscript entitled “Steroidogenic enzymes and steroid receptor expression in the accessory sex glands of the horse” (Manuscript ID; animals-1286059 ) was describes the expression level and distribution of steroids receptors and steroidogenic enzymes in the accessory sex glands of the horse evaluating by real-time PCR and immunohistochemistry. Results demonstrated that sex steroid receptors (AR, ER-a, ER-b) were expressed in all accessory sex glands throughout all stage of life in male horse, however the expression of steroidogenic enzymes were weak. The manuscript showed basic information for understanding of accessory glands regulation and development by steroid receptors. There are some points needing corrections in the manuscript. Please consider the suggested edits listed below.

Query#1. Throughout the manuscript: Please consistent the abbreviation word of 3β-Hydroxysteroid dehydrogenase/Δ5-4 isomerase.

Response: This has been corrected so that all abbreviations are 3βHSD.

Query#2.Abstract: I could not understand which results were mentioned about the results of mRNA or protein. Please clarify it.

Response: Due to space constraints, the abstract focuses on mRNA expression.

Query#3. L33: "ER1" to "ESR1"

Response: Thank you, this has been corrected.

Query#4. Introduction and qPCR analysis: Please add the words of each abbreviation of gene and protein when each words appear at first.

Response: We have added abbreviations or full names where they were missing.

Query#5. L70: Please delete [7-9].

Response: It has been fixed as suggested.

Query#6. Tissue collection: In this study each groups' replicates were three. Therefore, please add the individual's month or age of each horse. Also, I wonder that are there any differences of expression level of steroid receptors among age in each group? 

Response:  We appreciate the suggestion; unfortunate we did not have enough replicates within groups to answer whether age was a factor. Age of all animals have been specifically included as suggested.  

Query#7. Statistical analysis: Did you analyze the normal distribution by Shapiro-Wilk test and homogeneity of variance by Bartlett's test of each data? If so, please describe it. When the normality and homogeneity of variance of data are rejected by Shapiro-Wilk test and/or Bartlett's test, you have to analyze the data by Kruskal-Wallis test. 

Response: Yes, and this has been added to the methods.

Reviewer 2 Report

The manuscript is well written, and the information provided is new and valuable; therefore, it is acceptable to publish. Some minor errors listed below should be corrected.

1. The first paragraph (Line 41 ­­­­­­­­­­­­­­­­­­­­­­­­- 54) in “Introduction” can be deleted.

2. The primer sequences of AR in Table 1 should be added ref.

3. Line 202: “……Masson’s Table 280.days gestation),” is omitted some words or is error. Please correct.

4. The format of Table 2 can be improved with the addition of box heads.

5. In Fig. 4 and 5, the plates are repeated, and the denotations of samples are inconsistent. Please select the suitable one only in each Fig.

6. Line 285: “plan a role” please change to “play a role”.

7. Line 288: “vary between life stages” please change to “vary among life stages”.

8. The numbers of references in "Rferences" are repeated. Please correct.

Author Response

Responses to Reviewer #2

The manuscript is well written, and the information provided is new and valuable; therefore, it is acceptable to publish. Some minor errors listed below should be corrected.

Query#1. The first paragraph (Line 41 ­­­­­­­­­­­­­­­­­­­­­­­­- 54) in “Introduction” can be deleted.

Response: Given the differences in glands between species, we felt a brief description of the horse glands is relevant! 

Query#2. The primer sequences of AR in Table 1 should be added ref.

Response: This reference has been added.

Query#3. Line 202: “……Masson’s Table 280.days gestation),” is omitted some words or is error. Please correct.

Response: This typo has been corrected.

Query#4. The format of Table 2 can be improved with the addition of box heads.

Response:  We have centered the Receptor names.

Query#5. In Fig. 4 and 5, the plates are repeated, and the denotations of samples are inconsistent. Please select the suitable one only in each Fig.

Response: We have removed the duplicated figures.

Query#6. Line 285: “plan a role” please change to “play a role”.

Response: This has been corrected!

Query#7. Line 288: “vary between life stages” please change to “vary among life stages”.

Response: This has been changed as requested.

Query#8. The numbers of references in "Rferences" are repeated. Please correct.

Response:  This has been corrected.

Reviewer 3 Report

The manuscript focusses on an important topic, and is overall well written. The conclusions are well supported by the results and the discussion.

Some points that need attention are:

  • Title: maybe "receptors" rather than receptor
  • The simple summary is, in my opinion too generic. It has to be easy to understand, but still provide more info regarding the study itself
  • The abstract is too long. Please revise it according to the Journal’s guidelines
  • M&M: the authors should better describe the sampling protocol. This is important for reproducibility and transparency of reporting. How was euthanasia performed? What about the fetuses? Were tissue minced before incubation in RNA later? Was RNA later removed before freezing the samples?
  • Statistical analyses: I believe that there is a better way to cite R software, including the used version
  • Please, refer figures in the text according to the Journal’s guidelines
  • Check figure quality. Calibration bars are hard to read. Figure 3 is too big and some ANOVA letters have a red line underneath. I would suggest either spitting Fig 3 into more figures (maybe one for gene, highlighting only the genes that statistically change) or moving it to supplementary. What do error bars refer to? Figure 4-5-6: quality should be improved.

Author Response

Responses to Reviewer #3

The manuscript focusses on an important topic, and is overall well written. The conclusions are well supported by the results and the discussion. Some points that need attention are:

Query#1. Title: maybe "receptors" rather than receptor. Response: We have opted to remove the "s" from steroidogenic enzymes.

Query#2.The simple summary is, in my opinion too generic. It has to be easy to understand, but still provide more info regarding the study itself Response: The authors have included the receptors and enzymes being assessed to the simple summary.

Query#3. The abstract is too long. Please revise it according to the Journal’s guidelines Response: The abstract has been shorted to be meet Journal recommendations.

Query#4. M&M: the authors should better describe the sampling protocol. This is important for reproducibility and transparency of reporting. How was euthanasia performed? What about the fetuses? Were tissue minced before incubation in RNA later? Was RNA later removed before freezing the samples? Response: These details on euthanasia and collection have been added to the materials and methods section.

Query#5. Statistical analyses: I believe that there is a better way to cite R software, including the used version. Response: This has been cited as requested.

Query#6. Please, refer figures in the text according to the Journal’s guidelines. Response: This has been corrected.

Query#7.Check figure quality. Calibration bars are hard to read. Figure 3 is too big and some ANOVA letters have a red line underneath. I would suggest either spitting Fig 3 into more figures (maybe one for gene, highlighting only the genes that statistically change) or moving it to supplementary. What do error bars refer to? Figure 4-5-6: quality should be improved.  Response:  We have removed the duplicated poor quality figures 4 and 5.  We have fixed the red lines in figure 3, and removed CYP 17 and CYP 19 as those did not change in any gland.